# Challenges in the Integration of Omics and Non-Omics Data

**DOI:** 10.3390/genes10030238

**Published:** 2019-03-20

**Authors:** Evangelina López de Maturana, Lola Alonso, Pablo Alarcón, Isabel Adoración Martín-Antoniano, Silvia Pineda, Lucas Piorno, M. Luz Calle, Núria Malats

**Affiliations:** 1Genetic and Molecular Epidemiology Group, Spanish National Cancer Research Centre (CNIO), and CIBERONC, Melchor Fernández Almagro 3, 28029 Madrid, Spain; melopezdm@cnio.es (E.L.d.M.); lalonso@cnio.es (L.A.); pabloalarconmoreno@gmail.com (P.A.); iamartin@ceu.es (I.A.M.-A.); spineda@ext.cnio.es (S.P.); lucasp01012002@gmail.com (L.P.); 2Biosciences Department, University of Vic—Central University of Catalonia, Carrer de la Laura 13, 08570 Vic, Spain

**Keywords:** data integration, omics data, genomics, RNA expression, non-omics data, clinical data, epidemiological data, challenges, integrative analytics, joint modeling

## Abstract

Omics data integration is already a reality. However, few omics-based algorithms show enough predictive ability to be implemented into clinics or public health domains. Clinical/epidemiological data tend to explain most of the variation of health-related traits, and its joint modeling with omics data is crucial to increase the algorithm’s predictive ability. Only a small number of published studies performed a “real” integration of omics and non-omics (OnO) data, mainly to predict cancer outcomes. Challenges in OnO data integration regard the nature and heterogeneity of non-omics data, the possibility of integrating large-scale non-omics data with high-throughput omics data, the relationship between OnO data (i.e., ascertainment bias), the presence of interactions, the fairness of the models, and the presence of subphenotypes. These challenges demand the development and application of new analysis strategies to integrate OnO data. In this contribution we discuss different attempts of OnO data integration in clinical and epidemiological studies. Most of the reviewed papers considered only one type of omics data set, mainly RNA expression data. All selected papers incorporated non-omics data in a low-dimensionality fashion. The integrative strategies used in the identified papers adopted three modeling methods: Independent, conditional, and joint modeling. This review presents, discusses, and proposes integrative analytical strategies towards OnO data integration.

## 1. Introduction

Most health-related traits are complex in nature. They result from the interaction of multiple internal features/alterations with multiple external conditions over a lifespan [1]. Understanding these complex systems requires modeling exhaustive and appropriate data that characterizes in detail such features and conditions.

Big data in the biomedical field may refer to different scenarios encompassing large numbers of clinical (e-medical/e-health records, EMR/EHR) and epidemiological registries (hereinafter, non-omics data), as well as large biomarker datasets characterizing biological features, such as genomics, transcriptomics, proteomics, metabolomics, and metagenomics, among others. The latter type of data are commonly named omics data. While non-omics data are usually obtained through a pre-elaborated process done either by the subject when s/he reports on her/his life-style habits or symptoms, or by the physician/pathologist when s/he evaluates the characteristics of the disease or the tumor, omics data are generated by high-throughput biotechnological platforms delivering hundreds of thousands of raw (non-elaborated) variables. Recently, imaging-based high-throughput data is also generated and named radiomics.

Omics data integration has been addressed in recent years by several important reviews [2,3,4], and integrative efforts have been successfully conducted with already available examples of studies that integrated ≥ 2 different omics sets [5,6,7]. However, only a few of them resulted in omics-based algorithms with enough, though still controversial, predictive ability to be implemented into clinics or public health domains [8,9]. The relatively poor predictive ability of genomic data may partly be explained by the large variation of health-related traits explained by non-omics data, such as clinical and epidemiological variables [10]. Therefore, it is crucial to integrate omics and non-omics (OnO) data in the same models. This provides the opportunity to get insights into biological systems of health and disease. Unquestionably, this endeavor poses several challenges regarding data generation, capture, curation, sharing, analysis, visualization, as well as information privacy and storage.

What does OnO data integration mean in the biomedical arena? While it certainly refers to the inclusion and analysis of these two types of data in the same model/algorithm, several scenarios can be contemplated according to the number of each considered data type. There is no doubt that modeling > 1 omics data sets with > 1 non-omics variables falls under this integrative concept. However, should we consider integration when one omics data set (i.e., genome) is jointly modelled with only one non-omics variable (i.e., age or tumor stage)? In this scenario, the boundaries of the integrative picture become blurred and the definition depends on the purpose of the analysis and whether the inclusion of the non-omics variables aims only to control for a potential confounding effect or whether its prediction ability is being assessed in combination with the omics data. As a consequence of this confusion, the benefit of models including OnO data, is still unclear. This supports the need for a thorough dissection of the field to diagnose the challenges of the OnO data integrative endeavor and to identify the analytical strategies to reduce the variability of the study results.

In this review, we focus on the integration of OnO data to investigate complex traits, including disease risk and prognosis, according to the definition provided above. We first outline and examine the challenges of integrating the two types of data, we then present the integrative analytical strategies available, we describe the integrative attempts published in the literature, and we further propose statistical methods to be used in the analysis of OnO integrative models before concluding.

## 2. Challenges in Integrating Omics and Non-Omics Data

In this section, we focus mainly on the challenges of OnO data integration which are primarily related to the nature of both types of data and to the relationship between them, since much attention has already been paid to the integration challenges of only-omics data in previous reviews [3,4].

### 2.1. Challenges Due to the Nature of Non-Omics Data

#### 2.1.1. Non-Omics Data Are Complex and Heterogeneously and Subjectively Defined

There is an increasing awareness of the need for standards for non-omics data to integrate them in both predictive and inference models. Epidemiological data are subject to a survey mode, survey question standardization, and also context, which may influence data quality and comparability, and ultimately, the contribution of these variables in the outcome prediction. Standards are yet to be adopted in epidemiological data generated by different scientists or organizations through different procedures (i.e., questionnaires) to provide uniformity and consistency in this type of data, which may help scientists and data analysts to better use, share, and integrate them.

Clinical variables may also be affected by the complexity of their definition. A tumor stage, for instance, results from a combination of pathology and imaging information. Regarding clinical standardization, there are some initiatives as CDISC (Clinical Data Interchange Standards Consortium, http://www.cdisc.org) that harmonizes definitions and develops standards across the clinical space (i.e., the Study Data Tabulation Model (SDTM) and the Analysis Data Model (ADaM)) to enable information systems’ interoperability to improve medical research and related areas of healthcare.

Another important challenge relates to the nature of the aforementioned types of non-omics data, because they are subjective assessments that result from a complex elaboration process based on skills and previous knowledge of the evaluator which may lead to reporting biases (i.e., grading/staging, clinical decisions, or reporting past occupational exposures). In this regard, non-omics assessments totally differ from omics variables that are completely homogeneous and standardized data within the same data set. Integrating these different types of data poses challenges in the analytics strategy since the transformation or weighting of data may be required.

#### 2.1.2. Heterogeneity Across Non-Omics Data

The lack of uniformity of non-omics data, including qualitative and quantitative variables measured with different scales even to characterize a unique trait/exposure, also limits their integration in an OnO model to predict the outcome of interest and imposes both a conceptual challenge and a hurdle in practical data analysis. Moreover, data transformation (i.e., integrating variables with zero values) and data normalization procedures may be necessary prior to integration analysis, to avoid getting biased parameter estimates when the normality assumption required by some methods is violated.

#### 2.1.3. Large Scale Non-Omics Data

To date, the inclusion of non-omics data into integrative prediction models has been at a low dimension. However, the hype generated by so-called Big Data has also affected the healthcare industry. The advent of Big Data in the clinical setting has increased by the availability of EHRs (e-health records), unstructured medical text, and image data. These “large in scale, high in dimension” non-omics data, along with the design of well-characterized large and longitudinal epidemiological studies at an unprecedented scale, has led to the need for the integration of high dimension non-omics data in models. The use of other digital data sources coming from different wearable devices, such as smart watches, wristbands or wearable health equipment, are also expected to revolutionize epidemiology. The availability of longitudinal data concerning vital signs or environmental variables is expected to shed light on the knowledge of disease dynamics [11]. In addition to the high volume of data, other challenges of using digital epidemiology data are related to the collection, mining, access (i.e., limited and costly access), and data sharing (i.e., variability in definition/standardization of variables and subjective filters applied to the raw data which are needed to analyze those data).

The high dimensionality in non-omics data also implies the presence of (1) correlation structure between these variables, (2) large scale longitudinal data, (3) data sparseness (i.e., medications, laboratory or diagnosis tests), and (4) data missingness, which in contrast to omics data, are not independent on the participating individuals. In this regard, multi-dimensional approaches need samples with all the OnO data measured in the same individuals. All of these aspects must be taken into account in integration models.

Moreover, the advent of using EHRs will also be challenging in processing both objective and subjective traits, as well as structured and unstructured data. Subjective traits were defined by Jette as phenotypes that the “physician cannot assess directly with confidence and have to rely on patient (i.e., pain, physical, social, and emotional function) [12]. On the contrary, objective outcomes are those which the “physician can assess directly with confidence” [13]. Unstructured data, as the physician notes, which are in many cases embedded within semi-structured EHR data, are the most frequent data in the medical records. Although they have been mostly ignored, they are needed to understand the whole of a patient, and it will be needed to process and utilize them.

### 2.2. Challenges Due to the Relationship between Non-Omics and Omics Data

#### 2.2.1. Ascertainment Bias

In a case-control design, the integration of OnO data may be affected by the presence of ascertainment bias. In this type of epidemiological design, individuals are enriched for the risk factors of the study. If omics data are generated on the basis of the subject’s exposure, ascertainment could induce additional correlation between all OnO data [14]. It is known that omics profiles are not independent of demographic factors [15]. For example, age and gender may be associated with DNA methylation values [16,17]. In the clinical setting, genomic variables may be correlated with clinical variables due to population stratification [18]. Furthermore, when survival is the outcome, an insufficient clinical follow-up and the larger incompleteness affecting the clinical variables, in contrast to the completeness of high-throughput molecular data, may bias the effect estimates of the remaining clinical variables in a greater manner than their counterparts of omics variables. On the contrary, traits identified from an observational resource, such as medical records, may also be subject to the presence of ascertainment bias, since the probability that a particular phenotype is recorded is not uniform across patients or diseases.

#### 2.2.2. Interactions between Omics and Non-Omics Data

In order to understand the underlying mechanisms of the disease of interest, it is important to consider the combined interactions between the factors included in the model, irrespectively of their nature (omics vs. non-omics). The interaction between data types can be complex as well: gene expression changes may imply phenotypic abnormalities, and this results in a more complex relationship between molecular and clinical data.

### 2.3. Other Challenges

#### 2.3.1. Fairness

According to Van de Geer, a fair model is a model where all variable blocks, each block representing a set of variables sharing similar characteristics, contribute equally, in contrast to a model dominated by only a few of the different sets [19]. In OnO integrative modeling, should each variable or block contribute equally to the outcome? How can we prevent clinical variables from being penalized when combined with a high-throughput dataset?

#### 2.3.2. Presence of Subphenotypes

The consideration of heterogeneous phenotypes in the model may also add complexity to the OnO model definition. However, ignoring the presence of subphenotypes may affect the performance of the OnO model [20].

## 3. Integrative Analytical Strategies

The strategies for building hybrid models that contain both omics and non-omics data can be classified as: Independent modelling, conditional modelling, and joint modelling (Figure 1). While the joint modeling strategy is the most proper integrative approach, independent and conditional modeling are also commonly used approaches to jointly model OnO data.

### 3.1. Independent Modeling Approach

This strategy, also known as late integration, implies that both the omics and the non-omics data models are built independently [21]. The non-omics data model is built independently of the omics variables by fitting a model that only includes clinical/epidemiological variables or already well-established risk or prognostic score/factors identified and reported in previous efforts. In parallel, the omics variables are selected by considering a model only including omics variables. Both modelling processes typically require variable selection or dimension reduction. The independently selected omics and non-omics variables are then combined in a final model. The predictive accuracy of the combined model is compared with that of the non-omics data model.

Although independent modeling is the simplest integrative approach and, probably, the most common strategy for combining OnO data, this approach cannot capture the correlation/interaction structure of the datasets of different natures. To overcome this limitation, Nevins et al. [22] and Pittman et al. [23] proposed tree-based approaches to combine clinical and molecular scores in such a way possible interactions among OnO data are considered. Whether this approach is also applicable to omics data should be elucidated. Another caveat of the independent modeling strategy is that the predictive power of omics data tends to be overestimated since the trait is also used in the feature selection process.

### 3.2. Conditional Modeling Approach

This strategy consists in first defining a clinical model with non-omics variables and second, adding omics variables to the already built non-omics model. In other words, in the conditional modelling approach, the selection of omics variables is performed by considering a model that contains or adjusts for the previously selected clinical/epidemiological covariates. The key point of this conditional modeling approach is to decide which omics variables should be added to the clinical model. There are different ways to implement this strategy, the simplest one, though not recommended, is univariate selection, where each omics variable is tested individually and added to the clinical model if there is an increase in the prediction accuracy. As discussed in Bovelstad et al. [24], univariate selection performs poorly, usually yielding worse predictions than the clinical model approach. A more powerful strategy is to perform partial dimension reduction, which consists in considering the joint model with all omics and clinical variables and applies a dimension reduction process only to omics variables. One of such dimension reduction approach is least squares-partial least squares (LS-PLS) [25]. The major caveat of this method is that it suffers from convergence problems, and its performance also depends on the level of collinearity between the two types of data. Alternative approaches to LS-PLS, when the outcome is binary, are partial least square regression [26], ridge regression [27], and LASSO [28] performing dimensionality reduction only on omics data [24]. Other approaches for time dependent variables are described in [29,30]. Binder et al. proposed the algorithm CoxBoost which implements Cox penalized regression that allows some covariates (clinical variables) to be unpenalized [29] and Li et al. applied partial dimension reduction of the supergenes identified after estimating principal components with the omics variables, meanwhile considering the clinical covariates [30]. A common drawback of all the above methods is that they are computationally intensive.

### 3.3. Joint Modeling Approach

Under this strategy, omics and non-omics data are jointly modelled in a supervised or unsupervised manner. While there is a growing body of articles on multi-marker and multi-omics data integration [2,4,7,20,31,32], the literature that explicitly addresses how to integrate omics and non-omics data in a joint modeling approach is scarce. Following Ritchie’s suggestion [3], we can further classify the joint modeling approaches of OnO data into multi-staged (i.e., separate analysis of the associations between the different data types and subsequently with the outcome of interest) and meta-dimensional analyses (i.e., simultaneous analysis of the different data types). One of the first examples of a meta-dimensional approach is the study by Sun et al. that performs concatenation-based integration and joint variable selection of both OnO data using the i-relief algorithm [33]. Those classified as meta-dimensional analyses were further classified into three groups as concatenation-based integration, transformation-based integration, and model-based integration.

## 4. Attempts of OnO Data Integration in Clinical and Epidemiological Studies

We searched the PubMed electronic database using keywords to identify studies integrating OnO data towards their association with or prediction of the trait of interest, as well as to evaluate their joint classification performance. The search strategy included a combination of keywords related to omics, non-omics, and data integration, for the period 1 December 2009 to 1 October 2018. The logic terms used were: ((integration AND (risk OR score OR prediction OR prognosis) AND (epidemiological OR clinical OR environmental OR exposure) AND (genomic OR GWAS OR genetic OR transcriptomics OR proteomics OR metabolomics OR gene expression OR epigenomics OR epigenetic OR microbiome OR metagenomics))).

The search strategy generated 1,634 records. In this review, we only considered those articles integrating non-omics and high-throughput generated omics data sets in the modeling of the disease/trait as defined in the Introduction. The search resulted in a total of 16 studies almost all of them belonging to the cancer research area (see Table 1). We were first surprised by the small number of published studies at present that performed a “true” integration of OnO data. Although this contribution does not intend to be a systematic review, we consider that the identified papers constitute a representative sample of the attempts done in the field up to date. Hereinafter, we describe the objectives of the OnO integration, the outcomes and the OnO data types considered in the models, as well as the integrative analytical strategies applied in the selected papers (Table 1).

### 4.1. Study Objective

All selected papers aimed to evaluate the prediction performance of the OnO integrative models.

### 4.2. Study Outcome

Garali et al. identified OnO variables discriminating cases with spinocerebral ataxia from controls [37]. Only two out the 16 selected studies integrated OnO data to predict the risk of skin [45], and bladder [40] cancers. The rest of papers integrating OnO data analyzed cancer outcomes. Among the cancers analyzed were breast [24,35,38,41,43,44], central nervous system [24,25,34], liver [36], hematological [24], melanoma [39], bladder [20], kidney [42], and several cancers [46]. Six studies integrated both data types to evaluate the ability to predict the survival time [24,35,36,42,43,46]. Four studies transformed the survival time into a binary outcome (i.e., survival at a given time) [35,39,41,44]. López de Maturana et al. [20] transformed each time to event into several binary outcomes by accounting for censoring and time. Two studies analyzed the logarithm of survival time also accounting for censoring [34,38] . And two studies assessed the treatment prediction response as a categorical variable: Responders vs. non-responders [25,43].

### 4.3. Omics Data

Most of the papers only integrated one type of omics data [20,24,25,36,40,43,44,45]; five papers integrated two omics data types [35,38,41,42,43]; and four papers integrated > 2 omics data [34,36,39,46]. Gene expression data was the most commonly used high-dimensional omics data [24,25,34,35,36,38,39,42,43,44,46] followed by copy number alterations (CNA) [25,34,35,38,41,46], and SNPs [20,34,40,45,46]. Methylation data was considered by five selected papers [34,36,42,43,46] and three studies integrated microRNA (miRNA) data [36,39,46]. Only Jayawardana et al. [39] integrated protein expression data and Garali et al. [37] integrated 754 metabolite biomarkers in a predictive model. In those studies that integrated > 1 omics data set, gene expression was the most informative type in terms of prognostic utility [39,46], followed by microRNAs, and DNA methylation profiles [46].

### 4.4. Non-Omics Data

All the selected papers incorporated the non-omics data in a low-dimensionality fashion, meaning that only a few variables were integrated in the models. Non-omics information was a quite heterogeneous group of data formed by both categorical and continuous variables. The majority of non-omics data were clinico-pathological variables, including treatment, tumor stage, tumor size, lymph status, histological type, estrogen receptor status, progesterone receptor status or human epidermal growth factor receptor (see Table 1 for further details). Specific tumor scales, such as Breslow thickness and Clark’s level in melanoma [39] or classifications as Lauren classification in stomach adenocarcinoma [46] or the international prognostic index in lymphoma [24] were also used as non-omics clinico-pathological variables. Moreover, cancer subtype definition based on gene-expression signatures as PAM50 signature for breast cancer [42,43,46] as well as Mammaprint [46] were also considered. Jayawardana et al. were the only ones integrating metabolic imaging obtained by magnetic resonance spectroscopy, along with pons volume [39]. In addition, demographical data as age, gender, ethnicity, or region were also considered. Smoking status was the only epidemiological/life-style variable included in the risk models [40]. None of the papers considered large scale clinical or epidemiological data in their models.

### 4.5. Integrative Analytical Strategies

The integrative strategies used in the identified papers adopted the three different modeling methods described before (see Table 1): (1) Independent modeling, (2) conditional modeling, and (3) joint modeling, which were implemented using one-step or two-step designs. The published studies applied these methods assuming low-dimensional non-omics data, while omics data were high dimensional and required some variable selection, dimension reduction, or regularization process.

Examples applying the independent modeling approach are found in [36,39,42,43,44,46]. Thompson and Marsit [42], combined both multistage and meta-dimensional elements in a Methylation-to-Expression Feature Model (M2EFM) by first defining a molecular score that combined DNA methylation and gene expression and then performing a second regression to integrate clinical variables in a prognostic model for clear cell kidney cancer. van Vliet et al. [44] determined the optimal sets of features from each data type separately by using different classifiers such as the nearest mean, the simple Bayes, the 3-Nearest-Neighbor, the support vector machine, and the Tree Classifier. They then used these sets of features and all training samples in the final integrative model. Jayawardana et al. [39] used multiple types of omics data (i.e., microRNA, mRNA and protein expression) to integrate them with clinico-pathological variables also using an independent modeling approach. Briefly, they selected an optimized set of omics features integrated as a molecular signature of each data type (known as pre-validated vector) and then modelled them in combination with the clinico-pathological data, creating a combined prognostic signature. Chaudhary et al. [36] applied a transformation-based integration of multi-omics data independently from the clinical variables by using a deep-learning approach to integrate RNA-seq, miRNA-seq, and DNA methylation data to identify subgroups of hepatocellular carcinoma. Zhu et al. [46] did similarly, which led to substantially improved prognostic performance over the use of clinical variables alone in half of the cancer types examined. Particularly, they used the kernel-fusion Cox model as the multi-omics kernel learning method for prognostic prediction. Their approach consisted of three steps: (1) They built a kernel reflecting the similarity of the individuals based on each omics data including mRNA, miRNA, CNA, methylation and mutational status; (2) they applied a kernel alignment approach to evaluate whether the similarity matrix built using an omics data set aligned well with its counterpart defined by another omics data type; and (3) they evaluated the prognostic performance of the molecular profile of each individual, which was assumed to follow a multivariate normal distribution with mean zero and (co)variance matrix *K* corresponding to a fused kernel. This resulted from the linear combination or fusion of each omics similarity matrices (somatic mutation, mRNA, miRNA, methylation, and copy number profiles), along with the clinical prognostic score and the polygenic risk score based on odds ratios reported in the literature. Through this way, prognosis-relevant signals from multiple pathways and involving a large number of omics biomarkers became visible only when aggregated. Zhu et al. applied, by far, the most comprehensive integrative approach [46].

Two studies applied the conditional modeling approach. Bazzoli and Lambert-Lacroix [25] adopted it using a one-step approach. They adapted the Least Squares—Partial Least Squares (LS-PLS) procedure to accommodate logistic regression hybrid models resulting into three different approaches: LS-PLS-IRLS (where IRLS denotes Iteratively Reweighted Least Squares algorithm), R-LS-PLS, and IR-LS-PLS differing in the way PLS is used in the classification context. The three approaches involved the incorporation of PLS scores resulting from the application of PLS regression on omics data into the OLS equations in an iterative way to obtain a one-step hybrid model accommodating OnO data. Bovelstad et al. [24] proposed a Cox regression model including OnO variables and applying different methods for dimensionality reduction only to omics data and found that the improvement of the OnO model varied among diseases: Whereas large improvements were obtained when OnO model was applied to diffuse large B-cell lymphoma (DLBCL) and neuroblastoma datasets, similar performance was obtained using gene expression data only vs. the integrative model.

Joint modeling integration was the most commonly used approach by the identified studies. Particularly, the majority of the studies applied the transformation-based meta-dimensional analysis, which combined multiple data sets after transforming each data type into an intermediate form, such as a graph or a kernel matrix. Three studies applied Bayesian Reproducing Kernel Hilbert spaces regressions as a modeling framework able to incorporate clinical risk factors and high-dimensional omics profiles [34,38,45], González-Reymúndez et al. also assessed the interactions between OnO factors [38]. Seoane et al. proposed a multiple kernel learning strategy implementing feature selection separately for each data type and by pathway membership [41]. Examples of the concatenation-based integration meta-dimensional analyses are found in [20,24,35,40,44]. Boulesteix et al. [35] applied the IPF-LASSO, a penalized regression method that allows different penalty terms to the different layers of information, whereas López de Maturana et al. [20,40] implemented a Bayesian LASSO coupled threshold modeling with different priors imposed for OnO data. Bovelstad et al. [24] proposed a Cox regression model including OnO variables and applying different methods for dimensionality reduction only to omics data. Van Vliet et al. [44] applied five classifiers (nearest mean, the simple Bayes, the 3-nearest-neighbor, the support vector machine, and the tree classifier) concatenating the omics and clinical features.

In addition, Garali et al. implemented a regularized generalized canonical correlation analysis (RGCCA) and a sparse generalized canonical correlation analysis (SGCCA) model-based integration approaches, in which each data type is analyzed separately and then combined in a final integrative model [37]. Rather than operating sequentially on parts of the measurements, this integrative approach aims at summarizing the relevant information between and within blocks of variables. Particularly, RGCCA incorporates a variable selection procedure and SGCCA allows both the extraction of biomarkers and the reduction of the multiblock datasets into a few meaningful components.

### 4.6. OnO Data Integrative Models Performance

The performance of the models considered in the selected papers was retrieved, whenever provided, and is displayed in Table 1. In general, the selected papers showed that the OnO data integrative models perform better in terms of classification performance than the only-clinical/epidemiological or only-omics model [24,25,34,38,39,41,42,43,44,46]. However, there were studies reporting no/slight improvement in terms of classification performance of OnO data integrative models [20,24,36,40,46]. The variability in terms of predictive improvement observed when applying OnO modeling could depend on different factors, such as the outcome, the omics and clinical/epidemiological variables, and the integrative method implemented. For example, Bovelstad et al. [24] found that the improvement of OnO model varied among diseases: While large model performance improvements were obtained when OnO data integration was applied to DLBCL and neuroblastoma datasets, no gain in performance was observed when gene expression data was integrated with clinic-pathological variables in the breast cancer dataset. Furthermore, the SNPs performed poorly in the outcome prediction across cancer types [20,40,46].

## 5. Recommended Integration Strategies

As previously discussed, we distinguished three different strategies for building hybrid models containing both omics and non-omics data: Independent modeling, conditional modeling, and joint modeling. Selected papers have applied these approaches to the integration of low-dimensional non-omics data and high dimensional omics data, which requires some variable selection, dimensionality reduction or regularization process before or during their modeling. However, these integrative modeling strategies also apply to high dimensional non-omics data, a scenario that is becoming more frequent because of new technological advances that constantly increase our capacity for obtaining additional information from many different sources (e.g., EHRs or wearable sensors).

Joint modeling approaches, where omics and non-omics data are jointly modelled in a supervised or unsupervised manner, are those recommended to integrate both large-scale OnO data, because they account for the correlation structure between the two data types and capture a larger complexity than the conditional or independent modeling. The decision of which modeling strategy (multi-staged or meta-dimensional) to follow should be done in accordance with the main objective of the analysis: Association testing or risk prediction [47]. Multi-staged analysis that models the relationship between the different layers of information will probably be preferable when the interest is to increase our biological knowledge of the disease mechanisms. On the other hand, the meta-dimensional approach will be more suitable when the goal is to improve prediction or prognosis for personalized medicine and modeling the mechanisms is not so relevant, although they are not exclusive for these purposes. Concatenation-based integration combines the different data types into a joint data matrix and performs a variable selection or dimension reduction to the whole data set. The concatenation approach cannot ignore that the different data types are expected to have different relevance to the outcome and the joint analysis should take this into account.

In addition to the methods used by the identified studies described previously, the following modeling strategies could be considered in jointly modeling OnO data. The kernel-fusion Cox model used in Zhu et al. [46] initially designed as multi-omics kernel learning method could be extended to include also non-omics data in a kernel reflecting also the similarity between the profiles for each multimodal data. iCluster and iCluster2 are examples of a model-based integration strategy and could also accommodate non-omics variables [6,48]. Briefly, they perform a joint latent variable model-based clustering method, where the latent component connects the different data specific models, inducing dependencies across the different data types. Furthermore, deep-learning methods could also be used in a model-based integrative approach [49,50]. Another machine learning approach, the tensor factorization, allows the integration of multiple data modalities and supports dimensionality reduction and identification of latent groups [51]. A tensor factorization is a multidimensional array where each modality spans one axis and helps identifying group-wise interaction. Since it is an unsupervised method, it may be used to identify phenotypes, as it has been done in the Multi-Ethnic Study of Atherosclerosis (MESA) for discovering subgroups of heart failure patients. A drawback of this method is the interpretability of the results.

## 6. Concluding Remarks

Disentangling a complex trait requires not only the understanding of its “complex” biological system but also the combinatorial effects of other factors (i.e., host-related, environmental, socio-economics, etc.). The integration of OnO data can lead to finding new risk factors of a disease, propose better predictive models, distinguish patients with favorable response to treatment, and therefore help in the future of personalized medicine [52]. Unfortunately, OnO data integrative efforts are still scarce, although they are expected to become more frequent because of the advent of Big Data in the medical field.

In general, integrating both molecular and clinical data results in better prognostic models than either type alone as has been shown by several authors [39,43,44,46]. Possible explanations are that individual classifiers collect associations with the outcome of interest and their redundancy leads to a better prediction; that the clinical set of features adds some additional information which is not captured by the omics data; and that relevant signals may come from multiple pathways and involve a large number of omics biomarkers, the effect of which may be visible only when aggregated. However, model improvement has not always been observed when OnO data is integrated [36,40].

In any case, exploring OnO data integration becomes a must in the biomedical field. It requires method development, validation, and standardization. This review represents an endeavor towards these aims by identifying the challenges that OnO data integration presents, as well as discussing and proposing integrative analytical strategies. We hope it guides OnO data integrative efforts.

## Figures and Tables

**Figure 1 genes-10-00238-f001:**
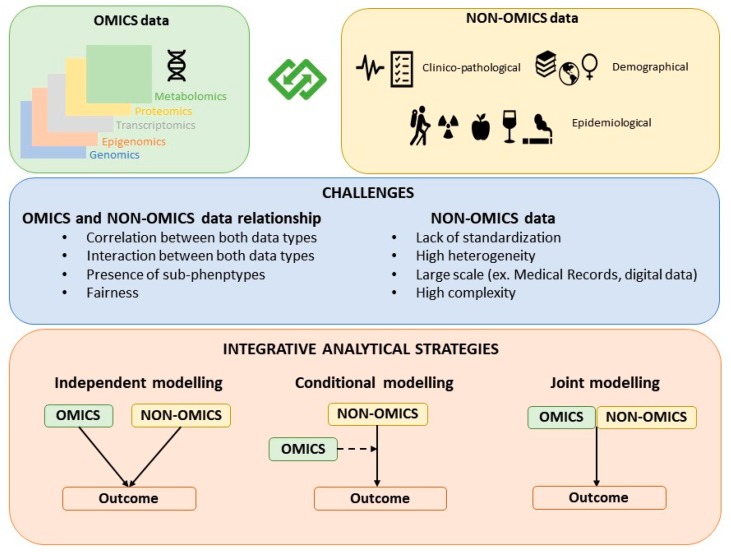
Classification of the strategies for building OnO models.

**Table 1 genes-10-00238-t001:** Main features of the identified studies conducting omics and non-omics data integration.

Reference	Title	Outcome	Big Data: Omics and Image Data	Non-Omics	Objective	Model Performance	Approach
[25]	Classification based on extensions of LS-PLS using logistic regression: application to clinical and multiple genomic data	Dataset1: Response of childhood malignant embryonal tumors of the CNS to therapy Dataset2: ER status in breast cancer	Dataset1: gene expression data Dataset2: somatic CNA	Dataset1: sex, age, chemo CX, chemo VP. Dataset2: grade, tumor stage, HER2 status, tumor size, progesterone receptor status	Dataset1: Prediction performance: AUC Misclassification rates Dataset2: predict ER stratification of a novel breast tumor to select appropriate treatment: AUC Misclassification rates	Dataset1: AUC_non-omics_ = 0.60 AUC_omics_ = 0.92 AUC_OnO_ = 0.82−0.90 Dataset2: AUC_non-omics_ = 0.87 AUC_omics_ = 0.84 AUC_OnO_ = 0.93	Conditional modeling
[34]	Whole-genome multi-omic study of survival in patients with glioblastoma multiform	Survival time in glioblastoma	TCGA: SNP + methylation + CNV + gene expression	TCGA: Sex + use of temozolomide	Predictive ability: AUC	AUC_non-omics_ = 0.71 AUC_omics_ = non-provided AUC_OnO_ = 0.72	Joint modeling
[24]	Survival prediction from clinico-genomic models—a comparative study	BrCa dataset: Survival time DLBCL dataset: Survival time Neuroblastoma dataset: Survival time	BrCa dataset: gene expression DLBCL dataset: gene expression Neuroblastoma dataset: microarray gene expression	BrCa dataset: tumor diameter, lymph node status, grade and NPI DLBCL dataset: International Prognostic Index (IPI) Neuroblastoma dataset: NB2004 stratification index	Prediction performance: deviance	Breast cancer dataset: DiD_non-omics_ = −12.5 DiD_omics_ = −14 to −2 DiD_OnO_ = −14 to −10 DLBCL dataset: DiD_non-omics_ = −12.5 DiD_omics_ = 0 to −9 DiD_OnO_ =−8 to−19 Neuroblastoma dataset: DiD_non-omics_ = −42 DiD_omics_ = −28 to −45 DiD_OnO_ =−40 to −50	Conditional and joint modeling
[35]	IPF-LASSO: Integrative L_1_-penalized regression with penalty factors for prediction based on multi-omics data	AML dataset: OS BrCa dataset: Distant relapse free survival time Pathological response (binary)	AML dataset (TCGA): microarray gene expression somatic CNA BrCa dataset: microarray gene expression	AML dataset (TCGA): age, % blast cells in bone marrow, white blood cell count per mm^3^, and sex BrCa dataset: age, nodal status, tumor size, grade, estrogen receptor, and progesterone receptor	Predictive ability: Prediction error curves, Brier score, integrated Brier score	AML dataset: IBS_non-omics_ = non-provided IBS_omics_ = non-provided IBS_OnO_ = 0.211–0.196 Breast cancer dataset: IBS_non-omics_ = non-provided IBS_omics_ = non-provided IBS_OnO_ = 0.134–0.127	Joint modeling
[36]	Deep learning based multi-omics integration robustly predicts survival in liver cancer	TCGA dataset: Survival LIRI-JP cohort: Survival NCI cohort: Survival Chinese cohort: Survival E-TABM-36 cohort: Survival Hawaiian cohort: Survival	TCGA dataset: RNA-seq, miRNA-seq, DNA methylation LIRI-JP cohort: RNA-seq NCI cohort: microarray gene expression Chinese cohort: miRNA E-TABM-36 cohort: gene expression Hawaiian cohort: DNA methylation	TCGA dataset: Stage, grade, race, gender, age, and risk factor	Predictive ability: C-index, Brier score Long-rank *p*-value	LIRI-JP^a^: C-index_non-omics_ = 0.55 C-index_omics_ = 0.75 C-index_OnO_ = 0.74 NCI ^a^: C-index_non-omics_ = 0.45 C-index_omics_ = 0.67 C-index_OnO_ = 0.65	Independent modeling
[37]	A strategy for multimodal data integration: application to biomarkers identification in spinocerebellar ataxia	SCA dataset: SCA subtypes and controls	SCA dataset: 754 metabolites	SCA dataset: MRS of the cerebellum, calorimetry information, volume of the pons	Graphical Reliability of parameter estimates % of times a variable has a non-null weight	Non-provided	Joint modeling
[38]	Prediction of years of life after diagnosis of breast cancer using omics and omic-by-treatment interactions	METABRIC dataset: Log (survival time)	METABRIC dataset: CNV, gene expression (GE) Also interactions with treatment: GExCT, GExRT, GExHT	METABRIC dataset: Age, cancer subtype, histological type, Nottingham Prognostic Index (tumor size, grade and nodal involvement), treatment	Prediction accuracy: AUC % of variance explained Definition of two groups: high and low risk	AUC_non-omics_ = 0.72–0.77 AUC_omics_ = non-provided AUC_OnO_ = 0.74–0.81 ^b^	Joint modeling
[39]	Determination of prognosis in metastatic melanoma through integration of clinic-pathologic, mutation, mRNA, microRNA, and protein information	MIA dataset: Survival grouped into good prognosis and poor prognosis	MIA dataset: mRNA, somatic mutation, microRNA expression, protein expression	MIA dataset: Nineteen clinical/pathological data: ulceration and thickness of primary tumor, number of lymph nodes with metastases, and size of nodal metastasis at the time of staging and others	Prediction error rate (ER) Kaplan-Meier curves	ER_non-omics_ = 30% ER_mRNA_ = 25% ER_protein_ = 35% ER_microRNA_ = 37% ER_OnO_ = 29–33%	Independent modeling
[40]	Whole Genome Prediction of Bladder Cancer Risk With the Bayesian LASSO	SBC/EPICURO dataset: BC risk	EPICURO/SBC dataset: SNP	SBC/EPICURO dataset: age + gender + region + smoking	Prediction: AUC	AUC_non-omics_ = 0.65 AUC_omics_ = 0.53 AUC_OnO_ = 0.65	Joint modeling
[20]	Prediction of non-muscle invasive bladder cancer outcomes assessed by innovative multimarker prognostic models	SBC/EPICURO dataset: Time to first recurrence (TFR) Time to progression (TP)	EPICURO/SBC dataset: SNP	SBC/EPICURO dataset: TFR: Area + gender + # of tumors + TSG + tumor size + treatment TP: Area + age + # of tumors + TSG + # of recurrences + treatment	Prediction: AUC, *R*^2^	TFR ^c^: AUC_non-omics_ = 0.62 AUC_omics_ = 0.55 AUC_OnO_ = 0.61	Joint modeling
[41]	A pathway based data integration framework for prediction of disease progression	METABRIC dataset: Survival vs. not survival at 2000 days	METABRIC dataset: Gene expression, CNV	METABRIC dataset: ER status only, disease &treatment group, grade of disease, stage, histological type, HER2 status, age, tumor size, NPI (tumor size, lymph node, grade), tumor cellularity, PAM50-based subtype	Accuracy	Acc_non-omics_ = non-provided Acc_omics_ = 0.64–0.71 Acc_OnO_ = 0.66–0.80	Joint modeling
[42]	A methylation-to-expression feature model for generating accurate prognostic risk scores and identifying disease targets in clear cell kidney cancer	TCGA dataset: OS in clear cell renal cell carcinoma	TCGA dataset: Gene expression (RNA seq), DNA methylation profile	TCGA dataset: Age, sex, tumor stage	Classification performance: *C*-index	C-index_non-omics_ = 0.776 C-index_omics_ = 0.702 C-index_OnO_ = 0.792	Independent modeling approach
[43]	Methylation-to-expression feature models of breast cancer accurately predict overall survival, distant-recurrence free survival, and pathologic complete response in multiple cohorts	TCGA dataset: OS (time) Terunuma dataset: OS (time) Kao dataset: OS (time) Hatzis1 dataset: Distant recurrence-free survival (time) Hatzis2 dataset: Distant recurrence-free survival (time) Pathologic complete response in BC (binary)	TCGA dataset: gene expression and methylation profiles Terunuma dataset: gene expression Kao dataset: gene expression Hatzis1 dataset: gene expression Hatzis2 dataset: gene expression	All datasets: AJCC stage, Age, ER status, PR status, HER2 status, PAM50-based subtype	Classification performance: *C*-index, AUC	OS TCGA ^d^: C-index_non-omics_ = 0.75 C-index_omics_ ^e^ = 0.69 C-index_OnO_ = 0.79	Independent modeling
[44]	Integration of Clinical and Gene Expression Data Has a Synergetic Effect on Predicting Breast Cancer Outcome	Vijver dataset: Poor/good outcome Other datasets: Poor/good outcome	Vijver dataset: Expression data Other datasets: Expression data	Vijver dataset: 45 clinical variables Otherdatasets: age, tumor size, grade, ER stats, lymph node, NPI	Classification performance: error rate, AUC	Vijver dataset: AUC_non-omics_ = 0.75 AUC_omics_ = 0.74 AUC_OnO_ = 0.74-0.78	Independent and Joint modeling
[45]	A Comprehensive Genetic Approach for Improving Prediction of Skin Cancer Risk in Humans	Kreger’s dataset: Skin cancer risk	Kreger’s dataset: SNPs	Kreger’s dataset: Age, ethnicity	Classification performance: AUC	AUC_non-omics_ = 0.53–0.54 AUC_omics_ = 0.63–0.64 AUC_OnO_ = non-provided	Joint modeling
[46]	Integrating Clinical and Multiple Omics Data for Prognostic Assessment across Human Cancers	TCGA datasets: Survival prediction in multiple cancer types	TCGA datasets: Significant SNPs (PRS, as fixed effect), somatic mutation, mRNA, miRNA, methylation, copy number, immune and metagenes signatures	TCGA datasets: Age, stage, Lauren classification (STAD) Also depends on the cancer type MammaPrint and PAM50 gene signatures	Classification performance: *C*-index	Ovarian and HNSC ^f^: C-index_non-omics_ = 0.60; 0.61 C-index_omics_ = 0.61; 0.61 C-index_OnO_ = 0.64;0.64	Independent modeling

CNS: Central Nervous System; ER: Estrogen receptor; NPI: Nottingham Prognostic Index; NB2004: German neuroblastoma trial; OS: Overall Survival; AML: Acute myeloid leukaemia; CT: Chemotherapy; RT: Radiotherapy; CT: Chemotherapy; TSG: Tumor stage and grade; PRS: Polygenic risk score; HNSC: Head and neck squamous cell carcinoma; DLBCL: Diffuse large B-cell lymphoma; IBS: Integrated Brier score. ^a^ Models performance in the largest datasets. ^b^ It corresponds to the AUC of COV+GE+GExHT model. ^c^ No improvement in classification performance was also obtained in TP. ^d^ We provide only the results for OS, when no external validation was considered. Similar performances were obtained when the external validation was performed. ^e^ Performance of M2EFM Meth+Exp model. ^f^ We report the C-index results for the cancers where the largest prognostic power was achieved.

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
