# Peer review of "Challenges in the Integration of Omics and Non-Omics Data"

_genes, 2019, doi:10.3390/genes10030238_

Round 1

Reviewer 1 Report

This is a review of methods that combine omics and non-omics data for the purpose of predicting a clinical outcome. The authors identify a set of 16 papers relevant to their review. They classify papers for the type of data used and the strategy for data integration (Section 2). The review is followed by some considerations about non-omics data (Section 3) and by an explanation of the methods used in integration (Section 4).

This is a very interesting approach to review the state of the art in data integration in predictive medicine that adds to previous reviews more focused on the power of multiomics methods. The paper does a good job in describing methodologies and caveats for data integration. However, after reading one feels that more could have been done to provide guidelines and indicate which approaches have worked more than others, and whether is really interesting or in which cases, to pursue an integrative study. The review could be greatly improved if authors go back to the papers and try to extract, when possible, quantitative data on the performance improvement obtained when clinical data is complemented by omics, or vice-versa, and which type of data added more predictive value. Some indication of this is given around line 115 and in the Concluding Remarks sections, but details are missing and hence the reader does not really get a good sense of the relevance of these integration strategies. These data could be presented in a tabular format

Moreover, the paper feels disconnected. First data and methods are described for the selected papers. Then problems associated to non-omics data are discussed without any connection with the previous section. For example, it would be interesting to use how to different issues of non-omics data are addressed and solved by the different integration strategies.

Then the paper goes and talks again in section 4 about Integrative designs and tools to explain, from a more theoretical perspective, each type of integration approach. A more logical structure of the paper is to join this section with the  “Integrative analytical strategies” part of section 2. As it is now, the information on Section 4 feels somehow redundant.

Moreover, I would suggest to make an effort to explain some of the mathematical concepts in lay terms. For example, when reading their definition of Conditional Modelling and Independent Modelling, not a clear picture of the differences between these two methods can be drawn, as apparently both start with non-omics data and then add the omics information to improve models. Maybe a diagram or cartoon could be used to illustrate the concepts.

Finally, I identified some grammatical errors, and there might be more. Please proof-check the English.

“they built a kernel reflecting the similarity of the individuals based on each omics data

including which includes mRNA, miRNA, CNA, methylation and mutational status”. Remove which includes

“Examples applying the independent modelling approach is found in…”  change is to are

Author Response

Reviewer 1:

This is a review of methods that combine omics and non-omics data for the purpose of predicting a clinical outcome. The authors identify a set of 16 papers relevant to their review. They classify papers for the type of data used and the strategy for data integration (Section 2). The review is followed by some considerations about non-omics data (Section 3) and by an explanation of the methods used in integration (Section 4).

This is a very interesting approach to review the state of the art in data integration in predictive medicine that adds to previous reviews more focused on the power of multiomics methods. The paper does a good job in describing methodologies and caveats for data integration. However, after reading one feels that more could have been done to provide guidelines and indicate which approaches have worked more than others, and whether is really interesting or in which cases, to pursue an integrative study.

1. The review could be greatly improved if authors go back to the papers and try to extract, when possible, quantitative data on the performance improvement obtained when clinical data is complemented by omics, or vice-versa, and which type of data added more predictive value. Some indication of this is given around line 115 and in the Concluding Remarks sections, but details are missing and hence the reader does not really get a good sense of the relevance of these integration strategies. These data could be presented in a tabular format.

Response: We thank the reviewer for her/his suggestion. We have included a new paragraph in 4.6. OnO data integrative models performance where we report on the potential improvement achieved when OnO data modelling strategies were considered in the selected papers. Moreover, we also comment why the improvement of OnO data integrative models is not always achieved depending on different factors such as the outcome, the omics and clinical/epidemiological variables, and the integrative method implemented.

2. Moreover, the paper feels disconnected. First data and methods are described for the selected papers. Then problems associated to non-omics data are discussed without any connection with the previous section. For example, it would be interesting to use how to different issues of non-omics data are addressed and solved by the different integration strategi

Then the paper goes and talks again in section 4 about Integrative designs and tools to explain, from a more theoretical perspective, each type of integration approach. A more logical structure of the paper is to join this section with the  “Integrative analytical strategies” part of section 2. As it is now, the information on Section 4 feels somehow redundant.

Response: Following the suggestion of the reviewer, we have restructured the paper. In the current version, the paper is organized as: 1. Introduction, 2. Challenges in integrating omics and non-omics data, 3. Integrative analytical strategies, 4. Attempts of OnO data integration in clinical and epidemiological studies, 5. Recommended integration strategies, and 6. Concluding remarks. In section 3. Integrative analytical strategies, an overview of the possible strategies is given and, in the section 4.5. Integrative analytical strategies, a more detailed description of the methods used in the selected studies is provided based on the strategies described in section 3.

3. Moreover, I would suggest to make an effort to explain some of the mathematical concepts in lay terms. For example, when reading their definition of Conditional Modelling and Independent Modelling, not a clear picture of the differences between these two methods can be drawn, as apparently both start with non-omics data and then add the omics information to improve models. Maybe a diagram or cartoon could be used to illustrate the concepts.

Response: We have restructured the paper to avoid redundancy and clarifying the integration strategies. The “Conditional modelling approach” (lines 273-282) paragraph has been extended to better explain this strategy and clarify the differences with the “Independent Modelling Approach”. Furthermore, Figure 1 has been modified accordingly to explain graphically each integrative modelling strategy.

4. Finally, I identified some grammatical errors, and there might be more. Please proof-check the English.

 “they built a kernel reflecting the similarity of the individuals based on each omics data

including which includes mRNA, miRNA, CNA, methylation and mutational status”. Remove which includes

 “Examples applying the independent modelling approach is found in…”  change is to are

Response: We have carefully reviewed the manuscript and correct typos and grammatical errors.

Reviewer 2 Report

The review paper summarizes recent studies that integrated omics (high throughput gene expression and related measurements) data with other, heterogeneous data types (epidemilogical, demographic variables etc). Computational and statistical challenges for integrating and interpreting these datasets are substantial, and so this is a useful review. It is surprising that there have been so few of what the authors deem "real" OnO integration efforts -- it would be interesting for the authors to comment on why this is the case and whether there are prospects for more thorough attempts at such integration.  My main comment is that the recommendations for integration strategies (Section 4.4) could be made sharper. Why is "joint modeling" preferred over conditional modeling?  This was not clearly explained or motivated.  

Author Response

REVIEWER 2:

1. The review paper summarizes recent studies that integrated omics (high throughput gene expression and related measurements) data with other, heterogeneous data types (epidemilogical, demographic variables etc). Computational and statistical challenges for integrating and interpreting these datasets are substantial, and so this is a useful review. It is surprising that there have been so few of what the authors deem "real" OnO integration efforts -- it would be interesting for the authors to comment on why this is the case and whether there are prospects for more thorough attempts at such integration.

Response: We attempted to stress this important point in several in several sections of the paper. In lines 182-186 and in lines 491-494, we comment on the fact that the advent of big data in the medical field is expected to increase the efforts of integrating OnO data, particularly to also accommodate high dimensional clinical/epidemiological data, for example those coming from EHR. We also emphasized this idea in the Concluding remarks section (see lines 547-549).

2. My main comment is that the recommendations for integration strategies (Section 4.4) could be made sharper. Why is "joint modeling" preferred over conditional modeling?  This was not clearly explained or motivated. 

Response: To clarify this issue, we extended the comments about the particularities of the conditional modelling approach in lines 273-282.

Round 2

Reviewer 1 Report

In my previous revision, I made four requests for improvement. I feel that the paper has improved in structure, but other concerns have been poorly addressed

I am sorry, but I still do not see a clear difference in they way they describe Independent and Conditional modelling. In both cases, they mention that  first a model is created with the non-omics variables alone, then a variable selection or dimension reduction is applied to the omics, and then the two are combined in a final model. What is the fundamental difference here? Is the difference that for the Conditional model only the omics data is added if it improved the prediction? Other criteria? Still confusing.

The authors added an extra paragraph to discuss improvements of OnO models over clinical models alone. This is nice, but this paragraph does not say much, other that in some cases it helps, in other not and it could be for all kind of reasons. If this is the conclusion, great, but I would like to see a more systematical evaluation here. I think that if they take table 1 and go study by study and indicate what the improvement was, readers would have more valuable information to look at. As it is now, it is not very useful.

English still needs revision. The sentence ..."the similarity of the individuals based on each omics data including which includes mRNA, miRNA, CNA" is still there. New weird sentences have been introduced at the revised text. For example, "On the other hand" in line 182 has no "On one hand" counterpart and it is simply not needed. "The advent of using EHRs" or "All of these aspects would be necessary to account for in the integration models" are other examples.

Author Response

In my previous revision, I made four requests for improvement. I feel that the paper has improved in structure, but other concerns have been poorly addressed

I am sorry, but I still do not see a clear difference in they way they describe Independent and Conditional modelling. In both cases, they mention that  first a model is created with the non-omics variables alone, then a variable selection or dimension reduction is applied to the omics, and then the two are combined in a final model. What is the fundamental difference here? Is the difference that for the Conditional model only the omics data is added if it improved the prediction? Other criteria? Still confusing.

R: The main difference between the two strategies is that in the independent modelling approach, the selection of both the omics and non-omics variables is done independently of each other. A model is first fit including only the outcome variable and the omics variables without any adjustment for non-omics covariates. The same process is done with the non-omics data by excluding the omics variables. In the final model, the selected omics variables are integrated with the selected non-omics ones. Instead, in the conditional modelling approach, the selection of the omics variables is performed by considering a model that already contains and adjusts for the previously selected non-omics covariates. In both approaches, the selection criteria could be in terms of association or in terms of prediction.

The authors added an extra paragraph to discuss improvements of OnO models over clinical models alone. This is nice, but this paragraph does not say much, other that in some cases it helps, in other not and it could be for all kind of reasons. If this is the conclusion, great, but I would like to see a more systematical evaluation here. I think that if they take table 1 and go study by study and indicate what the improvement was, readers would have more valuable information to look at. As it is now, it is not very useful.

R: Following the reviewer’s suggestion, an extra column with a summary of the models performance reported by each paper (whenever available) has been included in Table 1 of the reviewed version of the manuscript. Performing a systematic evaluation of the OnO integrative strategies is difficult because they have been applied to different datasets, each of them with different omics data types or outcomes. As commented in the paragraph 4.6, the variability in terms of predictive performance depends on several factors such as the outcome, the omics and clinical/epidemiological variables, and the integrative method implemented.

English still needs revision. The sentence ..."the similarity of the individuals based on each omics data including which includes mRNA, miRNA, CNA" is still there. New weird sentences have been introduced at the revised text. For example, "On the other hand" in line 182 has no "On one hand" counterpart and it is simply not needed. "The advent of using EHRs" or "All of these aspects would be necessary to account for in the integration models" are other examples.

R: We apologise for these language errors. We have revised again the manuscript and correcting the grammatical errors indicated by the reviewer as well as others independently identified. We hope the Ms is now suitable in this regard.

Round 3

Reviewer 1 Report

I am fine with this last version. Thank you for taking my siggestions into considerations